# Paranormal belief, psychopathological symptoms, and well-being: Latent profile analysis and longitudinal assessment of relationships

**Kenneth Graham Drinkwater** *, **Andrew Denovan**, **Neil Dagnall**

Faculty of Health and Education, Department of Psychology, Manchester Metropolitan University, Manchester, United Kingdom

☯ These authors contributed equally to this work.
* k.drinkwater@mmu.ac.uk

**Data Availability Statement:** All relevant data files are available from the figshare repository (Link: https://doi.org/10.6084/m9.figshare.23748870)'.

## Abstract

Within non-clinical samples the relationship between paranormal belief (PB) and well-being varies as a function of level of psychopathology. Accordingly, believers are best conceptualised as a heterogeneous set of sub-groups. The usefulness of previous findings has been restricted by conceptual methodological limitations. Specifically, overreliance on cross-sectional design, the assumption that believers constitute a homogeneous group, and consideration of direct effects. Acknowledging these limitations, the present study investigated whether profile membership derived from PB and psychopathology (schizotypy and manic-depressive experience) predicted well-being (i.e., stress, somatic complaints, life satisfaction and meaning in life) across time. Concurrently, analysis assessed the mediating effect of theoretically important variables (transliminality, happiness orientation, fearful and skeptical attitude). A sample of 1736 (*M*age = 52, range = 18 to 88; 883 females, 845 males, eight non-binary) completed self-report measures indexing study constructs across time points. Latent profile analysis at baseline, identified three sub-groups varying in level of PB and psychopathology at baseline: Profile 1, moderate PB and high psychopathology; Profile 2, moderate PB and psychopathology; and Profile 3, moderate PB and low psychopathology. Path analysis demonstrated that Profile 1 (the highest psychopathology scoring profile) predicted higher negative and lower positive well-being over time in comparison with the other profiles. Moreover, Transliminality and Fearful Attitude positively mediated this relationship, whereas Skeptical Attitude produced negative mediation. These outcomes supported the presence of a sophisticated process underpinning the PB and well-being relationship. Overall, PB in the absence of psychopathology had no significant influence on well-being.

## Introduction

Paranormal belief (PB) persists within modern Western societies, with surveys reporting endorsement at around 50% [1, 2]. Although believers argue that there is evidence to support

**Funding:** ND and AD Application Grant Number: 25390 was approved by the Health, Psychology and Social Care Research Ethics and Governance Committee. The Bial Foundation: https://www.fundacaobial.com/com/ No, the funder had no role in study design, data collection and analysis, decision to publish, or preparation of the manuscript.

**Competing interests:** No. The authors have declared that no competing interests exist.

the existence of supernatural phenomena, critics contend that PB violates the known laws of science and lacks a robust empirical foundation [3]. PB is an important research topic because investigators have historically observed associations between supernatural credence and reduced well-being [4, 5]. These findings, however, have proved difficult to replicate because researchers have employed differing conceptualizations of PB [6–8].

Noting this, the present paper adopted Irwin's [6] operationalisation of PB as a proposition generated within the non-scientific community, which although not empirically verified to the satisfaction of the scientific establishment, is extensively endorsed by people who are normally capable of rational thought and reality testing. This classification is important from a psychological perspective because it regards PB as a form of delusional thinking present within non-clinical samples [9–12]. Commensurate with this view, reality testing deficits are a central feature of faulty reasoning [13–15] and psychosis [16]. The advantage of Irwin's [6] delineation is that it encapsulates construct breadth, while excluding idiosyncratic/developmentally immature beliefs and scientific hypotheses awaiting empirical appraisal.

Previous scholarly work examining relationships between PB and well-being has typically been limited by overreliance on cross-sectional design, the assumption that believers are an homogeneous group, and consideration of only direct effects. Regarding cross-sectional approaches, these are problematic because they assess data, exposure and outcome, at one point in time [17]. Consequently, analysis reflects a temporally constrained snapshot of complex processes [18], which prevents causal inferences, and is difficult to interpret. A concomitant issue when batteries of self-report instruments are employed is potential response bias (i.e., common method variance). This occurs when variance in answers is attributable to the method (measurement clustering) rather than the construct under observation [18].

Despite these limitations, researchers find cross-sectional designs attractive because they facilitate rapid collection of data. In this context, cross-sectional investigations are useful for assessing prevalence (i.e., the proportion of persons who endorse supernatural phenomena) and exploring relationships between variables. Findings can also inform hypothesis generation and research development. In terms of connections between PB and well-being, the inability of cross-sectional approaches to assess change over time limits their usefulness. Hence, articles frequently cite the need for complementary longitudinal studies [19–21]. Despite this, there remains an absence of studies using multiple time points.

With reference to paranormal believers constituting a homogeneous group, studies have demonstrated that this is not the case. Notably, Irwin [22] conducted a study examining belief typology. This involved analysis of data using hierarchical cluster analysis, which grouped similar responses. Irwin [22] found a four-cluster solution comprising traditional religious believers, tentative believers, skeptics, and new agers (endorsers of a diverse range of scientifically unsubstantiated beliefs and practices). A similar approach was employed by Schofield, Baker, Staples, and Sheffield [23], who used cluster analysis to group participants on religiosity, spirituality, and paranormal belief. Analysis identified four types of supernatural belief: believers, paranormal believers, skeptics, and religious believers. Differences between groups supported the notion that believers were able to distinguish between beliefs that possess equal empirical validity (Beck & Miller [24] refer to this as metaphysical chauvinism). For instance, whilst a religious person might endorse miracles, they may simultaneously reject the existence of paranormal phenomena such as psychokinesis (the ability to influence a physical system(s) using only psychic powers) [23].

These studies demonstrate that belief is heterogeneous and that believers are best conceptualised as sub-groups. Recent research using latent profile analysis (LPA) to combine PB with schizotypy and psychopathology-related factors (i.e., depression, and manic depressiveness) supports this notion [25, 26]. LPA is a categorical approach that uses theoretically selected

variables to identify latent subpopulations. Relatedly, LPA has identified differences within experiencers (as defined by breadth of paranormal encounters/involvement) that are related to variations in self-assessment of cognitive and neuropsychological capabilities [27, 28].

The ascription of uniformity stems from an overreliance on variable-centred analytical methods (i.e., correlation-based). This approach uses single construct measures and assumes that outcomes provide an estimate of associations among discrete variables averaged across the population [29]. Recognising this issue, the use of LPA to combine person-centred factors has advanced investigation by identifying differences between believers as a function of sub-group membership [25, 26]. This method builds on previous clustering work using schizotypy dimensions, which enhanced understanding of PB by highlighting the importance of subtle differences within believers [30–32].

The focus on cross-sectional designs assessing discrete factors also overemphasized the importance of direct effects. Although these provide useful insights, relationships are usually complex (i.e., influenced by other variables; [33]). Hence, as evidenced by relatively high levels of PB in non-clinical populations [1, 2]., PB alone is neither indicative of distress nor psychopathology [19, 34]. However, PB in the presence of other factors can influence well-being. Therefore, researchers need also to consider indirect effects (i.e., mediators and moderators). Illustratively, Thalbourne [35] proposed that transliminality, the "hypothe-sized tendency for psychological material to cross (trans) thresholds (limines) into or out of consciousness" ([36], p. 853), explained the relationship between PB and psychopathology. Explicitly, transliminality is a trait that denotes hypersensitivity to psychological material (i.e., ideation, imagery, affect, and perception) originating internally from the unconscious and/or the external world [37].

Consistent with this delineation, transliminality facilitates the transmission of information across the threshold between unconscious and consciousness. Enhanced permeability of psy-chological boundaries produces greater levels of unfiltered mentation and is associated with less conventional cognitions and perceptions such as higher levels of PB, mystical experience, positive attitudes to dream analysis, hyperaesthesia (increased sensitivity to sensory stimuli), creativity, and manic episodes. Correspondingly, Dagnall et al. [2] found that transliminality and psychopathology-related variables moderated the effect of PB on well-being. As levels of transliminality and unusual experiences (positive schizotypy) increased, the strength of the PB and perceived stress relationship increased. Furthermore, higher scores on transliminality, positive and disorganised schizotypy and manic and depressive experiences strengthened the relationship between PB and somatic complaints. Thus, PB was only allied to lower well-being when it interacted with transliminality and psychopathology-related variables.

Transliminality produces this effect by modifying sensory information flow [38]. High transliminality is indicative of a 'permeable' mental threshold or leaky filter [39], and loose sensory gating (neurological disinhibition) produces weak and/or erratic suppression of irrele-vant information, resulting in high cortical activation [40, 41]. Consequently, greater levels of material (unconscious, endo-psychic, and external) enter awareness [42]. PB in this context, provides an interpretative lens for classifying the ensuing anomalous sensations [43, 44]. This notion aligns with models that explain PB in terms of belief driven attributions, which propose that ambiguous phenomena and occurrences are defined and elucidated via established cre-dence, personal worldviews, and context [45, 46]. Congruent with this supposition translimin-ality is predictive of the general trait tendency to PB [47, 48].

Concomitantly, hypersensitivity to threatening stimuli is associated with susceptibility to psychosis, depression, and mania [49, 50]. The conceptual overlap between PB, transliminality, and psychopathology-related factors may explain why interactions between these constructs are related to negative well-being. Explicitly, increased exposure to spontaneous, poorly

attenuated sensory information concomitant with paranormal ideation increases the probability of experiencing heightened stress and unpleasant emotions [51].

Another factor that potentially influences the PB and well-being relationship is attitude. Haider [52] reported that PB increased orientation to happiness (i.e., pleasure in life), and that orientation to happiness enhanced satisfaction with life. A further factor that likely protects against adverse outcomes is skepticism. Believers demonstrate weak dispositional skepticism and low regard for the values of science [53]. Hence, greater levels of skepticism encourage individuals to question and critically assess the validity of their views. Likewise, when PB is allied to negative occurrences it is more likely to be associated with reduced well-being. Examples being adverse life events (e.g., developmental instability [54], childhood trauma, [55], and inadequate coping [2]).

### Present study

In response to previous study design limitations, the present study combined LPA with a multiple time point design and assessed whether changes in relationships between sub-groups were mediated by conceptually important factors. At baseline, this necessitated identification of profiles derived from PB and commonly correlated psychopathological constructs (i.e., schizotypy and manic-depressive experience). Then, across three time points, changes in well-being were assessed (i.e., somatic complaints, perceived stress, life satisfaction, and meaning in life). Selection of these outcomes derived from consideration of commonly used measures and were intended to index a broad range of physical and psychological characteristics. Mediating factors based on the previously outlined research were transliminality, fear of PB, skepticism, and orientation to happiness.

Given the complexity of the design and the exploratory nature of the study, precise hypotheses were not stated. However, it was expected that high levels of PB concurrent with greater levels of psychopathology would be related to lower well-being, and that transliminality would increase the strength of this relationship.

## Materials and methods

This study was a multiple time interval project, which was structured in a way to capture initial baseline information regarding paranormal belief and psychopathology, followed by potential mediating effects at subsequent time intervals. Use of multiple time points to test mediation is important to limit measurement bias, which can occur when assessing mediation within cross-sectional (i.e., single time point) research.

### Participants

In total, 1736 ($M$age = 52, range = 18 to 88) responded at four time points (scheduled at successive two-month intervals resulting in a study timescale of six months) from an initial sample of 4402 (61% dropout). This comprised 883 females ($M$age = 55, range = 18 to 88), 845 males ($M$age = 48, range = 18 to 80), and eight non-binary ($M$age = 45, range = 23 to 69). Recruitment used Bilendi, who obtain participants from an existing pool of individuals who possess an interest in research study involvement. A minimum age of 18 years was specified for the sample alongside a range of ages and equal genders. Online panel data is comparable with data that is sourced using conventional methods [56].

### Materials

All constructs were assessed using established, self-report instruments. Measurements occurred at four time points.

At time point 1 (baseline) participants completed scales assessing belief in the paranormal and psychopathology-related factors (schizotypy and manic-depressiveness).

**Belief in the paranormal.**   The Revised Paranormal Belief Scale [57] measured supernatural credence. The scale contains 26 items (e.g., 'Black magic really exists') accompanied by a 7-point Likert response format (strongly disagree to strongly agree) [6]. Higher scores indicate greater PB. The scale is valid and reliable [58, 59]. Observed internal reliability was satisfactory ($\alpha$ = .94).

**Schizotypy.**   The Oxford-Liverpool Inventory of Feelings and Experiences Short [60] measured schizotypy at Time 1. It is an abridged 43 item version of the O-LIFE [61] used with non-clinical samples. The instrument has four subscales. Unusual Experiences (12 items) assesses positive schizotypy (magical thinking, perceptual anomalies). Cognitive Disorganisation (11 items) measures disorganised elements of psychosis (e.g., poor concentration/attention). Introvertive Anhedonia (10 items) indexes negative schizotypy (avoidance of intimacy, withdrawal). Impulsive Non-Conformity (10 items) reflects self-control deficits, antisocial and impulsive behaviour. Items appear as questions (e.g., 'Is it hard for you to make decisions?') and respondents answer using 'yes/no'. Subscale reliability (alpha) ranges from .62 to .80 [60]. In the current investigation, reliability was comparable with previous research (Unusual Experiences $\alpha$ = .85, Cognitive Disorganisation $\alpha$ = .85, Introvertive Anhedonia $\alpha$ = .62, Impulsive Non-Conformity $\alpha$ = .64).

**Manic-depressiveness.**   The Manic-Depressiveness Scale [62] comprises two 9 item 'true/false' subscales: manic (e.g., 'My thoughts have sometimes come so quickly that I couldn't write them all down fast enough'), and depressive experience (e.g., 'I have experienced being so sad that I just sat (or lay in bed) doing nothing but feeling bad'). The scale has established validity and reliability [63]. Reliability estimates for Manic Experience ($\alpha$ = .62) and Depressive Experience ($\alpha$ = .79) were consistent with prior work [63].

At time points 2 (Transliminality) and 3 (Fearful and Skeptical Attitude, and Orientation to Happiness) the researchers assessed the effect of mediator variables.

**Transliminality.**   The Revised Transliminality Scale [39, 64] assessed propensity for psychological information to cross thresholds ('limines') into and out of consciousness. Items (e.g., 'Sometimes I experience things as if they were doubly real') appear as statements and participants respond using either 'yes or 'no'. Although 29 items are administered, 12 items are excluded due to gender and age biases [65]. The remaining 17 items form the Rasch version, which is psychometrically superior to the original iteration (i.e., demonstrates increased sensitivity and improved reliability) [65]. Higher scores reflect greater transliminality. In the current study, internal consistency was good ($\alpha$ = .87).

**Fearful attitude.**   The Fear subscale from the Anomalous Experiences Inventory [66] assessed apprehension of the paranormal. This comprises 6 items (e.g., 'Hearing about the paranormal or psychic experiences is scary'), and participants record their responses using 'true/false'. The instrument is valid and reliable [66]. In this study, internal reliability was good ($\alpha$ = .86).

**Skeptical attitude.**   Hurrt's Professional Skepticism Scale [67] evaluated the extent to which respondents critically assess evidence. The instrument comprises 30 context-free items and participants indicate the degree to which they endorse statement (e.g., 'I usually notice inconsistencies in explanations') using a 6-point Likert scale (1 = Strongly Disagree, to 6 = Strongly Agree). Internal consistency is typically good (i.e., .85 and .91) [67]. In this investigation, internal reliability was excellent ($\alpha$ = .90).

**Orientation to happiness.**   The Orientation to Happiness Scale [68] measures three pathways (life orientations) to well-being (i.e., pleasure, engagement and meaning). The instrument comprises 18 items (e.g., 'For me, the good life is the pleasurable life') presented as

statements. Participants respond via a 5-point Likert scale (1 = Very much unlike me, to 5 = Very much like me). The measure is psychometrically validated [68]. In this study, internal reliability was good ($\alpha$ = .88).

At time point 4 the well-being outcomes (Perceived Stress, Somatic Complaints, Life Satisfaction, and Meaning in Life) were assessed.

**Perceived stress.** The Perceived Stress Scale (PSS-10) [69] measures the degree of uncontrollability and unpredictability in an individual's life over the past month. The PSS-10 includes 10 items (e.g., 'How often have you felt nervous and stressed?'); respondents record answers on a 0 (never) to 4 (very often) Likert scale. The PSS-10 has satisfactory reliability and validity [70]. The scale demonstrated good internal consistency in this study ($\alpha$ = .86).

**Somatic complaints.** The Somatic Symptom Scale-8 (SSS-8) [71] assessed susceptibility to somatic complaints. The SSS-8 is an 8-item measure that evaluates the extent to which bodily concerns (e.g., 'Chest pain or shortness of breath') have affected respondents during the previous seven-days. Each item is accompanied by a 5-point Likert ranging from 0 (not at all) to 4 (very much). The SSS-8 has good internal reliability [71]. Good internal reliability was observed ($\alpha$ = .89).

**Life satisfaction.** The Satisfaction with Life Scale [72] measured global cognitive judgments of satisfaction with life. The instrument has 5 items (e.g., 'In most ways my life is close to my ideal'); participants respond using a seven-point Likert scale ranging from 1 (strongly disagree) to 7 (strongly agree). The SWLS possesses good internal consistency [72]. This was also observed in this study, $\alpha$ = .80.

**Meaning in life.** The Meaning in Life Questionnaire [73] assesses search for and presence of meaning in life (i.e., importance and reason). The scale consists of 10 items (e.g., 'I have discovered a satisfying life purpose'), and respondents answer via a 7-point Likert scale, ranging from 1 (Absolutely Untrue) to 7 (Absolutely True). The scale possesses good reliability [73]. Within this study, good reliability existed, $\alpha$ = .84.

## Procedure

Respondents completed measures at four intervals two months apart. This period was sufficient for well-being outcomes to change [74]. Respondents initially received an information sheet. This informed them that the study involved comparing responses across time points. Respondents provided written informed consent by ticking/clicking a box indicating that they understood the nature of the study and then continued with the online measures. Hence, an ID number was required to enable response matching (ID number deletion occurred once this was achieved). Exclusion criteria were participants had to be at least 18 years of age and free from a diagnosed mental illness. Eligible consenting respondents progressed to the online measures. The study involved participants responding to the self-report measures online, with a requirement to complete baseline (paranormal belief and psychopathology-related) scales at the first time point. For the second time point (two months later), participants were contacted and asked to complete the Revised Transliminality Scale. Time point 3 (a further two months later) necessitated completion of Fearful and Skeptical Attitude, and Orientation to Happiness measures, and time point 4 (a further two months later; six months from baseline) required completion of well-being scales.

To reduce potential common method variance the researchers employed procedural remedies [75]. Firstly, to create psychological distance between constructs, section instructions emphasized measure uniqueness. Secondly, to control for order effects, the presentation sequence of measures varied across respondents. Finally, to lessen evaluation apprehension and social desirability respondents were told that there were no right or wrong answers.

Respondents were instructed to read statements carefully, work at their own speed, and answer all items.

This article was produced as part of a large, multiple time-point project investigating relationships between cognitive-perceptual factors, paranormal belief, and wellbeing. This study was unique because it combined latent profile analysis with a multiple time point design to assess whether changes in relationships between sub-groups were mediated by conceptually important factors over time [76]. This was not possible within the general analytical framework. In this context, the paper was a significant contribution as the usefulness of previous findings has been restricted by the assumption that paranormal believers form a homogeneous group, the use of cross-sectional approaches, and analysis focusing on direct effects.

The current paper was distinct because of the analytical approach used. Relationships were scrutinised over a sustained period alongside tests of mediation. From this perspective, temporal precedence is a central notion when identifying/assessing mediating processes, and data collected at multiple time points is typically more suitable for unveiling underlying mechanisms than alternative sources (e.g., cross-sectional data) [77]. Although, this paper was exploratory in nature it provides a framework for subsequent investigation that will potentially lead to greater conceptual understanding of the factors under consideration. Explicitly, identify the conditions under which belief in the paranormal is non-adaptive vs. adaptive.

**Ethics statement.** The Manchester Metropolitan University Faculty of Health, Psychology and Social Care Ethics Committee awarded ethical approval (Project ID, 25390). Recruitment of participants was granted by the committee and commenced from 11/12/2020 until 01/08/2022.

## Results

### Analysis plan

Following data screening and consideration of zero-order correlations, latent profile analysis (LPA) using Mplus 7 [78] examined sub-group membership based on PB and psychopathology scores. LPA assesses observed individual responses and categorises these into unmeasured group affiliation [79]. To ascertain the optimal quantity of latent profiles, a likelihood-based test, information criteria, and classification quality were employed. The likelihood test, Lo-Mendel-Rubin Adjusted Likelihood Ratio test (LMR-A-LRT; Lo et al., [80], compared a k profile with a k-1 profile solution. A significant *p*-value supports a higher profile solution. Information criteria included Akaike Information Criterion (AIC; [81]), Bayesian Information Criterion (BIC; [82]), and Sample-Size Adjusted BIC (ssaBIC; [83]), with lower values suggesting superior fit. Entropy is a measure of classification quality, with values closer to 1 indicative of clearer classification [84].

Next, a path model with profiles included as predictors was tested. This explored relationships across multiple time points between features of PB, Schizotypy, Manic-Depressiveness, and well-being outcomes (specifically Perceived Stress [PSS], Somatic Complaints [SC], Life Satisfaction [LS], and Meaning in Life [ML]). Transliminality, attitude (i.e., fearful, and skeptical), and happiness orientation were examined as mediators.

Prior to model testing, correlations were assessed for multicollinearity. To assess model fit, Confirmatory Fit Index (CFI), Standardized Root-Mean-Square Residual (SRMR), and Root-Mean-Squared Error of Approximation (RMSEA) were considered. Acceptable data-fit values are CFI > .90, SRMR < .08, and RMSEA < .06 [85]. Analysis of indirect or mediating effects used bootstrapping (1000 resamples) to create 95% bias-corrected confidence intervals [86].

**Table 1. Descriptive statistics and intercorrelations among paranormal belief, schizotypy, and manic-depressive experience.**

| Variable | Mean | SD | 1 | 2 | 3 | 4 | 5 | 6 | 7 |
|---|---|---|---|---|---|---|---|---|---|
| 1. Paranormal Belief | 1.35 | .74 | | .19** | .13** | .02 | .18** | .14** | .15** |
| 2. Unusual Experiences | .27 | .26 | | | .58** | .02 | .45** | .52** | .49** |
| 3. Cognitive Disorganisation | .35 | .30 | | | | .21** | .49** | .49** | .59** |
| 4. Introvertive Anhedonia | .38 | .23 | | | | | .19** | .11** | .23** |
| 5. Impulsive Non-Conformity | .23 | .20 | | | | | | .49** | .57** |
| 6. Manic Experience | .33 | .22 | | | | | | | .67** |
| 7. Depressive Experience | .29 | .26 | | | | | | | |

Note.

* indicates $p < .05$

** indicates $p < .001$

## Data screening

Assessment of normality revealed skewness values between -3.0 and +3.0 [87]. Correlations among the variables used for LPA were within the low to moderate range, and below .8 reflecting a lack of multicollinearity [88] (Table 1).

## Main analysis

LPA identified PB and psychopathology-related variables (Schizotypy and Manic-Depressive Experience) subgroups. To determine the optimal number of profiles, LPA was tested on an increasing number of profiles (starting with two) until this was no longer justified (i.e., non-significant LMR-A-LRT concurrent with minimal change in AIC, BIS and ssaBIC). In the analyses for two to four profiles, the AIC, BIC and ssaBIC decreased with each additional profile; the LMR-A-LRT was significant in the three-profile model, meaningfully improving model fit vs. the two-profile solution (Table 2). Statistical criteria and pattern interpretability indicated that the three-profile solution was optimal (Fig 1).

Profile 1, 'Moderate PB and High Psychopathology' (11.4% of the sample), exhibited relatively high psychopathology scores and moderate PB (vs. scale averages/norms). Profile 2, 'Moderate PB and Psychopathology' (32.6% of the sample) demonstrated similar PB alongside lower psychopathology (vs. Profile 1). Profile 3, 'Moderate PB and Low Psychopathology' (56% of the sample) evidenced moderate PB and relatively lower psychopathology (vs. Profiles 1 and 2). A lack of variation existed in relation to PB level across the profiles. Indeed, Profile 1 exhibited a mean PB of 4.42, whereas a mean PB of 4.52 and 4.76 existed for Profile 2 and 3 respectively.

Path analysis tested predictive relationships between the profiles and well-being outcomes over six months. The model included Transliminality (measured at two months), Fearful and

**Table 2. Fit of latent profile models.**

| Model | AIC | BIC | ssaBIC | LMR-A | LMR-A p value | Entropy |
|---|---|---|---|---|---|---|
| 2-profile | 1284.73 | 1404.85 | 1334.96 | 3132.54 | < .001 | .88 |
| 3- profile | 690.99 | 854.78 | 759.48 | 599.69 | < .001 | .84 |
| 4- profile | 480.38 | 687.86 | 567.14 | 222.86 | .323 | .82 |

*Note*. AIC = Akaike Information Criterion; BIC = Bayesian Information Criterion; ssaBIC = sample-size adjusted BIC; LMR-A = Lo-Mendell-Rubin-adjusted likelihood ratio test

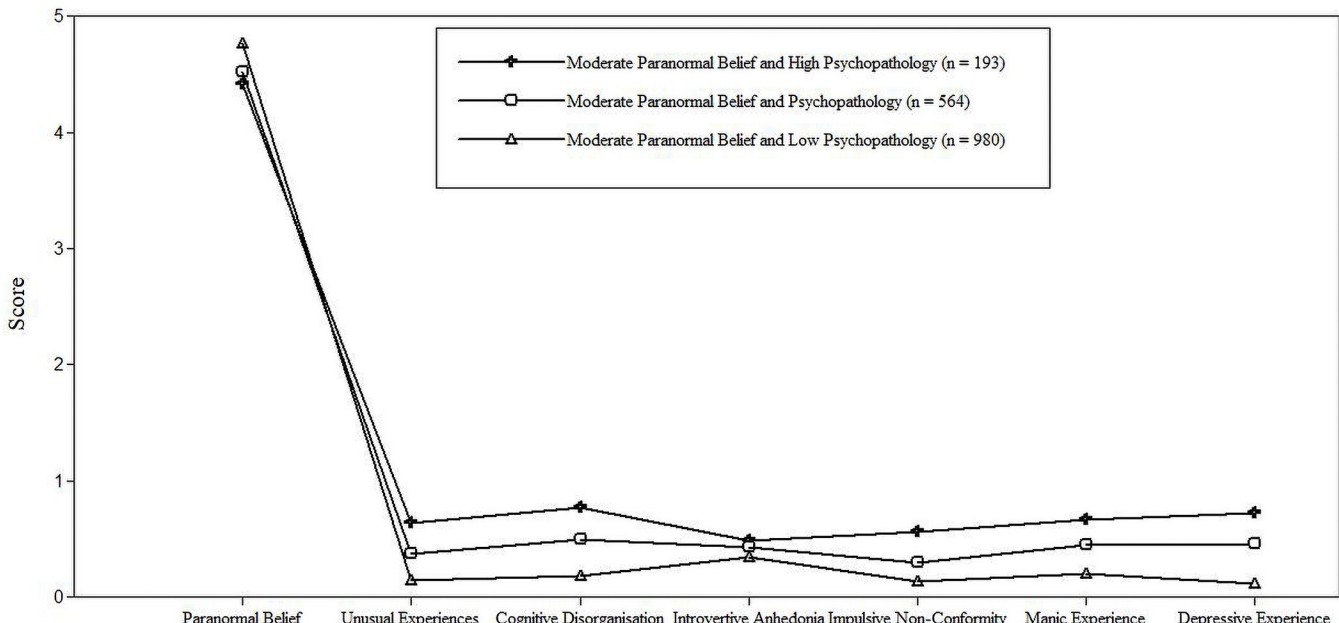

**Fig 1. Pattern of scaled mean scores for paranormal belief, schizotypy, and manic-depressive experience.**

Skeptical Attitude, and Orientation to Happiness (assessed at four months) as mediator variables. Profile 3 (lower scoring profile) was the reference group. Prior to the model test, correlations were assessed (Table 3). The model had good fit on all indices but RMSEA, $\chi^2$ (3, $N = 1736$) = 74.64, $p < .001$, CFI = .97, SRMR = .02, RMSEA = .11 (95% CI of .09 to .14).

Scrutiny of relationships using Full Information Maximum Likelihood revealed that Orientation to Happiness was not significantly related to either of the latent profiles or most outcomes (apart from Meaning in Life). Accordingly, a model was specified without this construct. This produced superior fit, $\chi^2$ (1, $N = 1736$) = 8.10, $p = .004$, CFI = .99, SRMR = .01, RMSEA = .06 (95% CI of .03 to .09). When compared with Profile 3, Profile 1 and 2

**Table 3. Correlations among latent profiles, transliminality, fearful attitude, skeptical attitude, orientation to happiness, and well-being outcomes.**

| Variable | 1 | 2 | 3 | 4 | 5 | 6 | 7 | 8 | 9 | 10 |
|---|---|---|---|---|---|---|---|---|---|---|
| 1. Profile 1 | | -.25** | .23** | .10** | -.16** | .06 | .30** | .19** | -.10** | .03 |
| 2. Profile 2 | | | .39** | .17** | -.18** | -.13** | .30** | .36** | -.01 | .07 |
| 3. Transliminality | | | | .23** | .06 | .27** | .29** | .44** | .01 | .05 |
| 4. Fearful Attitude | | | | | -.17** | -.11** | .18** | .20** | -.01 | -.04 |
| 5. Skeptical Attitude | | | | | | .19** | -.28** | -.15** | .20** | .15** |
| 6. Orientation to Happiness | | | | | | | -.02 | -.04 | .02 | .17** |
| 7. Perceived Stress | | | | | | | | .51** | .14** | .03 |
| 8. Somatic Complaints | | | | | | | | | -.04 | .07 |
| 9. Life Satisfaction | | | | | | | | | | .28** |
| 10. Meaning in Life | | | | | | | | | | |

*Note.* Profile 3 is the reference category

* indicates $p < .05$

** indicates $p < .001$

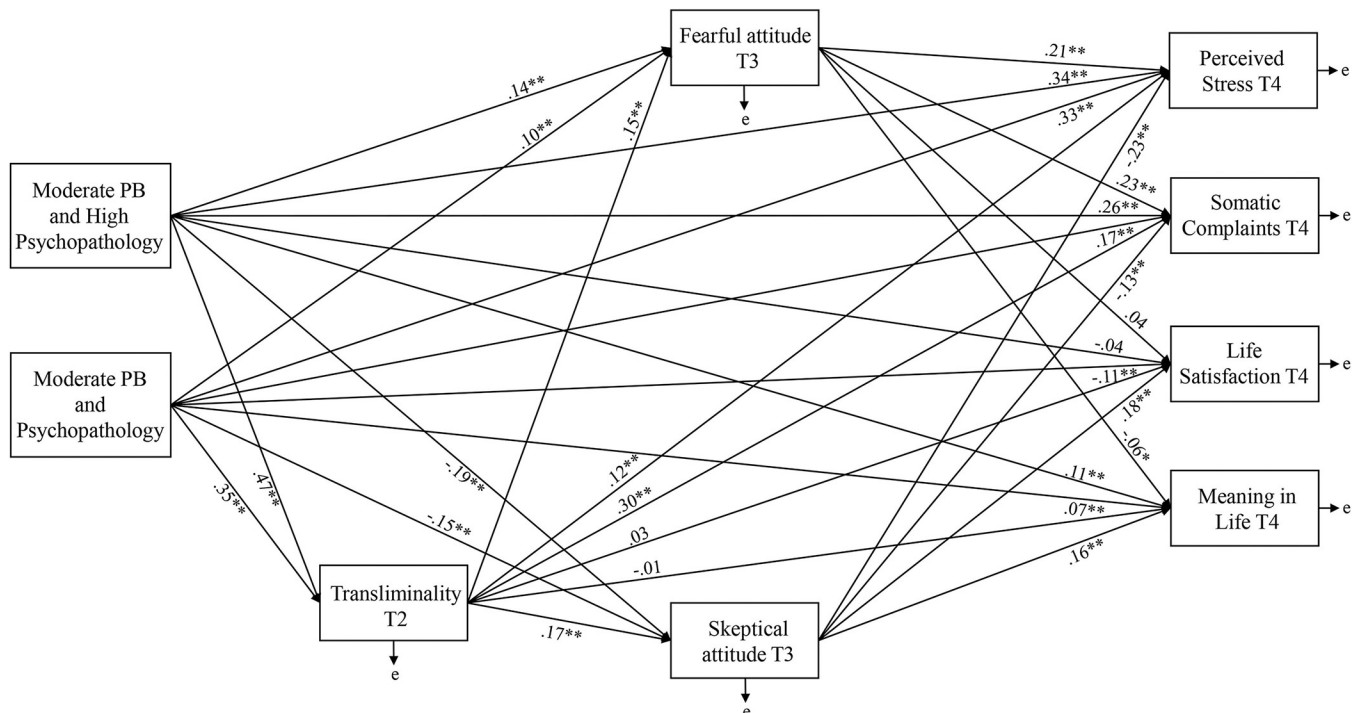

**Fig 2. Multiple time point mediation model depicting putative relationships between the latent profiles (reference category = Profile 3), transliminality, fearful attitude, skeptical attitude, and well-being outcomes.** *Note.* standardized regression weights between variables are shown. Error is not indicated but was specified for all variables. * $p < .05$, ** $p < .001$ using Bootstrapping significance estimates (1000 resamples). PB = Paranormal Belief.

significantly predicted greater PSS and SC, alongside lower LS and higher ML. Profile 1 (vs. Profile 2) evidenced comparably greater effects (Fig 2).

Transliminality and Fearful Attitude positively mediated the relationship in comparison with Profile 3, whereas Transliminality and Skeptical Attitude negatively mediated the relationship (Table 4). This reflects competitive mediation [89], which occurs when a mediated

**Table 4. Specific indirect effects of latent profiles on well-being outcomes through transliminality, fearful attitude, and sceptical attitude.**

| | Perceived Stress | Somatic Complaints | Life Satisfaction | Meaning in Life |
|---|---|---|---|---|
| Indirect path | $\beta$ (95%CI) | $\beta$ (95%CI) | $\beta$ (95%CI) | $\beta$ (95%CI) |
| Profile 1 > Transliminality | .13** (.07,.19) | .44** (.36,.53) | .05 (-.03,.12) | -.01 (-.09,.07) |
| Profile 1 > Fearful Attitude | .03* (.01,.04) | .03* (.01,.05) | .01 (-.01,.03) | -.03* (-.05,-.01) |
| Profile 1 > Skeptical Attitude | .14** (.10,.18) | .08** (.05,.11) | -.11** (-.15,-.07) | -.10** (-.13,-.06) |
| Profile 1 > Transliminality > Fearful Attitude | .01* (.01,.02) | .02* (.01,.02) | .00 (-.01,.01) | -.01* (-.02,-.01) |
| Profile 1 > Transliminality > Skeptical Attitude | -.06** (-.08,-.04) | -.03** (-.05,-.02) | .05** (.03,.07) | .04** (.02,.06) |
| Profile 2 > Transliminality | .06** (.03,.09) | .22** (.18,.26) | .02 (-.01,.06) | -.01 (-.04,.03) |
| Profile 2 > Fearful Attitude | .01* (.01,.02) | .01* (.01,.02) | .01 (-.01,.01) | -.01* (-.02,-.01) |
| Profile 2 > Skeptical Attitude | .07** (.05,.10) | .04** (.02,.06) | -.06** (-.08,-.04) | -.05** (-.07,-.03) |
| Profile 2 > Transliminality > Fearful Attitude | .01* (.01, .02) | .01* (.01, .02) | .00 (-.01,.01) | -.01* (-.02, -.01) |
| Profile 2 > Transliminality > Skeptical Attitude | -.03** (-.04, -.02) | -.02** (-.02, -.01) | .02** (.01, .03) | .02** (.01, .03) |

*Note.* Profile 3 is the reference category

* indicates $p < .05$

** indicates $p < .001$ using Bootstrapping significance estimates (1000 resamples)

and a direct effect exist in parallel, pointing in opposite directions. For LS, significant total and indirect effects did not exist. For ML, Profile 1 exhibited significant total and indirect effects, but a significant total effect did not occur for Profile 2. Skeptical Attitude positively mediated the relationship, and Transliminality and Fearful Attitude negatively mediated this in comparison with Profile 3.

## Discussion

Latent profile analysis (LPA) identified three sub-groups varying in level of PB and Psychopathology: Profile 1, moderate on PB and high on psychopathology; Profile 2, moderate PB and psychopathology; and Profile 3, moderate PB and low psychopathology. The emergence of believer profiles that differed in terms of level of psychopathology concurred with the notion that believers, like experiencers [27, 28], are best conceptualised as heterogeneous [25].

Identified profiles were conceptually important because PB was differently associated with well-being as a function of level of psychopathology [26]. Believers with high (Profile 1) and moderate (Profile 2) levels of psychopathology over time reported poorer well-being (higher PSS and SC, and lower LS and ML) than believers with low psychopathology (Profile 3). These relationships were stronger for Profile 1 (vs. Profile 2). This indicated that level of psychopathology rather than PB predicted lower well-being.

Broadly, these findings align with prior research, which reported that PB in the absence of cognitive-perceptual factors allied to psychopathology, was benign, or even in some instances beneficial to well-being [2]. For example, PB can perform adaptive psychological functions such as facilitate the development of self-concept [90], provide meaning [91], and create positive affect and reassurance [92–94].

Transliminality positively mediated the latent profile > negative well-being relationship. This effect was strongest for those with the most psychopathology (Profile 1 vs. Profile 3). Moreover, inclusion of Fearful Attitude positively mediated this relationship (negatively for positive well-being). Thus, when PB and psychopathology were high, Fearful Attitude and Transliminality predicted greater negative well-being. Inclusion of Skeptical Attitude concurrent with Transliminality produced competitive mediation (i.e., negative mediation of negative well-being, and positive mediation for positive well-being). Hence, when PB and psychopathology were high, Skeptical Attitude and Transliminality predicted lower negative well-being (higher stress and more somatic complaints) and greater positive well-being (increased life satisfaction and experience of meaning in life). These results suggest that attitude influences the extent to which PB in the presence of high psychopathology effects well-being.

The finding that transliminality mediates the relationship between PB and well-being concurred with [2]. This effect may occur because high levels manifest as the attenuated ability to actively suppress irrelevant information and increased awareness of psychological and physiological fluctuations [95–97]. The ensuing attention to spontaneous idiosyncratic mentation likely results in preoccupation with psychological and physical states, connectedness with experiences, and perceived lack of self-regulation and control (somatic complaints) and the environment (i.e., stress). This interpretation aligns with the observation that high (vs. low) transliminals report a range of undesirable outcomes (e.g., prescribed medication for a psychological condition and feeling overwhelmed) that are likely to adversely impact upon/or reflect issue with life satisfaction and well-being [96].

These suppositions require cautious interpretation for several reasons. Although profiles were statistically sound and represented potentially important variations in PB as a function of psychopathology, outcomes were descriptive and lacked a conceptual basis. Acknowledging this, the practical and clinical significance of the reported findings beyond the present study

are unclear [98]. Since the current article was exploratory and merely designed to establish whether indirect effects occurred over time, this was not problematic. However, from a general theoretical perspective further research is required to identity the specific psychopathology factors that most strongly influence the PB and well-being relationship.

As with recent work on paranormal experience-based profiles, further work is needed to establish sub-groups because LPA does not identify subtypes of individuals in the population. Instead, profiles derive from heterogeneity across variables within a model. Consequently, the number of emergent classes can vary across samples. To best establish equivalence, replication, and cross-validation methods (e.g., progressive elaboration, [99] are required). These prevent LPA misspecification by evaluating class stability and model fit [100]. This is an iterative process that will inform the development of conceptually driven sub-groups. Nonetheless, the emergent psychological profiles identified in this study advanced understanding by delineating the broad cognitive-perceptual factors that interact with PB (directly and indirectly) to affect well-being.

A further limitation was the lack of variability within PB scores. This prevented assessment of how differing levels of PB interacted with psychopathology-related variables. Accordingly, ensuing investigations should recruit samples with more diverse participants (i.e., both clinical and non-clinical) [101]. Additionally, further studies should employ scales that sample a broader range of unorthodox credence. These could include endorsement of scientifically unsubstantiated beliefs and pseudoscience [102].

Also, use of self-report measures can be problematic because there is a risk of participants over- or underestimating their beliefs, attitudes, and symptoms due to poor recall and/or social desirability bias. Finally, as a function of the exploratory nature of the study (and due to the high number of variables), total scores as opposed to subfactor scores were used. It would be useful for future studies to consider subfactor scores alongside total scores for measures including the Revised Paranormal Belief Scale, because research supports a bifactor structure [58].

## Author Contributions

**Conceptualization:** Kenneth Graham Drinkwater, Andrew Denovan, Neil Dagnall.

**Data curation:** Andrew Denovan.

**Formal analysis:** Andrew Denovan.

**Funding acquisition:** Andrew Denovan, Neil Dagnall.

**Investigation:** Kenneth Graham Drinkwater, Neil Dagnall.

**Methodology:** Kenneth Graham Drinkwater, Neil Dagnall.

**Validation:** Kenneth Graham Drinkwater.

**Writing – original draft:** Andrew Denovan, Neil Dagnall.

**Writing – review & editing:** Kenneth Graham Drinkwater, Neil Dagnall.

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
