## [Decision Letter · Decision Letter 0]

19 Nov 2023

PONE-D-23-24545Paranormal belief, psychopathological symptoms, and well-being: Latent profile analysis and longitudinal assessment of relationshipsPLOS ONE

Dear Dr. Drinkwater,

Thank you for submitting your manuscript to PLOS ONE. After careful consideration, we feel that it has merit but does not fully meet PLOS ONE’s publication criteria as it currently stands. Therefore, we invite you to submit a revised version of the manuscript that addresses the points raised during the review process.

We look forward to receiving your revised manuscript.

Kind regards,

Giulia Prete

Academic Editor

PLOS ONE

Journal Requirements:

Did you know that depositing data in a repository is associated with up to a 25% citation advantage (https://doi.org/10.1371/journal.pone.0230416)? If you’ve not already done so, consider depositing your raw data in a repository to ensure your work is read, appreciated and cited by the largest possible audience. You’ll also earn an Accessible Data icon on your published paper if you deposit your data in any participating repository (https://plos.org/open-science/open-data/#accessible-data).

5. Please upload a copy of Figure 1 and 2 to which you refer in your text on pages 32-33. If the figure is no longer to be included as part of the submission please remove all reference to it within the text.

6. We note you have included a table to which you do not refer in the text of your manuscript. Please ensure that you refer to Table 1 and 2 in your text; if accepted, production will need this reference to link the reader to the Table.

8. Please upload a copy of Supporting Information Table 1 & 2 which you refer to in your text on pages 14-15. 

**Additional Editor Comments:**

Both Reviewers stressed that it is a bit unexpected to define this study as a longitudinal study because different measures are collected across time. I agree with this idea and would like to invite you to better define this point, tigether with the other points raised by Reviewer 1. 

Reviewers' comments:

Reviewer's Responses to Questions

**Comments to the Author**

1. Is the manuscript technically sound, and do the data support the conclusions?

Reviewer #1: Yes

Reviewer #2: Yes

2. Has the statistical analysis been performed appropriately and rigorously? 

Reviewer #1: Yes

Reviewer #2: Yes

3. Have the authors made all data underlying the findings in their manuscript fully available?

Reviewer #1: Yes

Reviewer #2: Yes

4. Is the manuscript presented in an intelligible fashion and written in standard English?

Reviewer #1: Yes

Reviewer #2: Yes

5. Review Comments to the Author

Reviewer #1: • Authors are absolutely correct that most paranormal studies represent a single moment in time. There are virtually no studies that look at paranormal belief across time.

• Great introduction that covers the basis and details of “believers” and their brain function.

• State upfront that this study is “exploratory” and that “precise hypotheses cannot be stated”.

• Participants are self-enrolled via a website for people interested in participating in research of some kind. This could lead to some bias in the participants and will limit the extrapolation ability of this study.

• Reported the alpha values of the surveys and they were in a decent range showing good reliability on the surveys.

• I think that this paper strives to do a longitudinal study (across time) but they only take the PB survey at the beginning and not each time so the actual belief comparison doesn’t exist throughout the timeline, it is only a comparison of various psychological factors in relation to the original paranormal belief scale. However, if the authors can support that paranormal beliefs don’t really change all that much over the timeline of the study then all of my concerns will be met.

• Skeptical not Sceptical

• I found this paper interesting and a new and important piece of information concerning the type of believer personality is within humanity. However, the authors state that this is a longitudinal study, because it has surveys that are given over a time period of several months. However, none of the belief/psychology surveys were repeated so I have some reservations with calling this study a longitudinal study. Instead, this seems to be a study that has numerous qualifiers for various aspects of paranormal belief and psychoses that are associated with a preponderance of belief. In addition, this study links several of the mental factors to a person’s well-being. This is a neat, and unique aspect of this paper. I think it is a great statement that a person’s belief and their mental condition might work together to produce a terrible state of well-being. In contrast, it is great that some of the effects of believing in paranormal might actually offset some bad mental states. I would like to see this paper accepted, but to have it reworked to eliminate the longitudinal ideas and to have the paper reworked to be more about the connections of various aspects of mental state and paranormal belief.

o If this was truly a longitudinal study then the base scores (paranormal belief) would have been measured throughout the timeline, along with the other mental factors. It is possible for someone’s belief in paranormal subjects to change through time so it would be better to measure if there was any change in this (either all along the timeline, or at the beginning and end).

Reviewer #2: The present longitudinal study investigated whether the membership of profiles derived from PB and psychopathology predicts well-being over time, and it assessed the mediating effect of theoretically important variables (transliminality, happiness orientation, anxious and sceptical attitudes). Results supported the existence of a sophisticated process underpinning the relationship between PB and well-being. Overall, results indicated that PB, in absence of psychopathology, had no significant effect on well-being. "However, authors specify the exploratory nature of the study, so it is not possible to make specific hypotheses.

I would ask the authors to better clarify the concept of longitudinal study because none of the surveys were repeated in the different timepoints.

p. 18 close the parenthesis "(e.g., progressive elaboration, [94] are required".

6. PLOS authors have the option to publish the peer review history of their article (what does this mean?). If published, this will include your full peer review and any attached files.

Reviewer #1: No

Reviewer #2: No

---

## [Author Response · Author response to Decision Letter 0]

6 Dec 2023

General Comments

With formatting and referencing, we apologise that the submitted version of the manuscript did not adhere to the conventions. On checking, it appears that we used a version where the referencing did not wholly align with the numbering system in text. Additionally, within the reference section citations were a mixture of Vancouver and APA. This was due to the fact that the Authors use APA as a default and the conversion process was only partially implemented. The resubmitted manuscript contains the aligned, amended, and updated references as requested.

Reviewer #1:

Comment 

• Authors are absolutely correct that most paranormal studies represent a single moment in time. There are virtually no studies that look at paranormal belief across time.

Response

Thank you for acknowledging this point.

Comment

• Great introduction that covers the basis and details of “believers” and their brain function.

Response

We made every attempt to include a breadth of pertinent research and synthesise it around prevailing design, methodological, and analytical limitations.

Comment

• State upfront that this study is “exploratory” and that “precise hypotheses cannot be stated”.

Response

Yes – the approach due to novelty and complexity was exploratory in nature.

Comment

• Participants are self-enrolled via a website for people interested in participating in research of some kind. This could lead to some bias in the participants and will limit the extrapolation ability of this study.

Response

This is a common concern. We do check data quality and make comparisons with relevant studies. The provider has an established track record of providing data that is equivalent to that collected via traditional methods. In the context of my previous comments we are highly confident that these data were representative (more so than university-based samples) of general populations.

Comment

• Reported the alpha values of the surveys and they were in a decent range showing good reliability on the surveys.

Response

Reliability across measures was good as would be expected with a large general sample.

Comment

• I think that this paper strives to do a longitudinal study (across time) but they only take the PB survey at the beginning and not each time so the actual belief comparison doesn’t exist throughout the timeline, it is only a comparison of various psychological factors in relation to the original paranormal belief scale. However, if the authors can support that paranormal beliefs don’t really change all that much over the timeline of the study then all of my concerns will be met.

Response

This is a good/fair point. The study measures outcomes at various time points to see how these vary as a function of sub-group membership and hence is only partially longitudinal. Noting this we have changed the terminology throughout the paper. That stated, belief in the paranormal over the period measured is unlikely to change significantly. Indeed, we have previously reported that test–retest reliability is in the moderate to high range across time intervals. Also, that the RPBS demonstrates sustained internal consistency. These conclusions align with structural stability.

Comment

• Skeptical not Sceptical

Response

Have changed throughout. This works better in context, although there are generally cultural variations. We do agree though!

Comment

• I found this paper interesting and a new and important piece of information concerning the type of believer personality is within humanity. However, the authors state that this is a longitudinal study, because it has surveys that are given over a time period of several months. However, none of the belief/psychology surveys were repeated so I have some reservations with calling this study a longitudinal study. Instead, this seems to be a study that has numerous qualifiers for various aspects of paranormal belief and psychoses that are associated with a preponderance of belief.

Response

Agreed.

Comment

 In addition, this study links several of the mental factors to a person’s well-being. This is a neat, and unique aspect of this paper. I think it is a great statement that a person’s belief and their mental condition might work together to produce a terrible state of well-being. In contrast, it is great that some of the effects of believing in paranormal might actually offset some bad mental states. 

Response

Thank you – this was the primary motivation for examining the subject area. Also, to explore nuances and subtleties obscured by reductionist assumptions and resource limitations.

Comment

I would like to see this paper accepted, but to have it reworked to eliminate the longitudinal ideas and to have the paper reworked to be more about the connections of various aspects of mental state and paranormal belief.

Response

We have amended the references to longitudinal.

Comment

If this was truly a longitudinal study then the base scores (paranormal belief) would have been measured throughout the timeline, along with the other mental factors. It is possible for someone’s belief in paranormal subjects to change through time so it would be better to measure if there was any change in this (either all along the timeline, or at the beginning and end).

Response

Please see above.

Reviewer #2: 

Comment

The present longitudinal study investigated whether the membership of profiles derived from PB and psychopathology predicts well-being over time, and it assessed the mediating effect of theoretically important variables (transliminality, happiness orientation, anxious and sceptical attitudes). Results supported the existence of a sophisticated process underpinning the relationship between PB and well-being. Overall, results indicated that PB, in absence of psychopathology, had no significant effect on well-being. "However, authors specify the exploratory nature of the study, so it is not possible to make specific hypotheses.

I would ask the authors to better clarify the concept of longitudinal study because none of the surveys were repeated in the different timepoints.

Response

Thanks. These comments align with the other reviewer, and we have removed reference to longitudinal accordingly. 

Comment

p. 18 close the parenthesis "(e.g., progressive elaboration, [94] are required".

Response

Amended.

---

## [Decision Letter · Decision Letter 1]

4 Jan 2024

Paranormal belief, psychopathological symptoms, and well-being: Latent profile analysis and longitudinal assessment of relationships

PONE-D-23-24545R1

Dear Dr. Drinkwater,

We’re pleased to inform you that your manuscript has been judged scientifically suitable for publication and will be formally accepted for publication once it meets all outstanding technical requirements.

Kind regards,

Giulia Prete

Academic Editor

PLOS ONE

Additional Editor Comments (optional):

One reviewer stated that all previous issues were addressed, while the other did not agree to review the revised version of the manuscript. However, I have reviewed your manuscript myself and confirm that all issues raised have been satisfactorily addressed, so I am happy to accept the manuscript in its current form. 

Reviewers' comments:

Reviewer's Responses to Questions

**Comments to the Author**

1. If the authors have adequately addressed your comments raised in a previous round of review and you feel that this manuscript is now acceptable for publication, you may indicate that here to bypass the “Comments to the Author” section, enter your conflict of interest statement in the “Confidential to Editor” section, and submit your "Accept" recommendation.

Reviewer #1: All comments have been addressed

2. Is the manuscript technically sound, and do the data support the conclusions?

Reviewer #1: Yes

3. Has the statistical analysis been performed appropriately and rigorously? 

Reviewer #1: Yes

4. Have the authors made all data underlying the findings in their manuscript fully available?

Reviewer #1: Yes

5. Is the manuscript presented in an intelligible fashion and written in standard English?

Reviewer #1: Yes

6. Review Comments to the Author

Reviewer #1: • I still really like the idea of paranormal believers consisting of various types of believers, from religious believers to spiritual-type believers. I think it is very important to show that not all believers are the same and I think the current paper does a really good job of expressing (and supporting) this idea.

• The authors have done a good job of bringing down the language about this being a long-time study and also eliminating the references to any actual hypotheses. They are now very clear that this is an exploratory study and so the information within this paper should not be thought of as a conclusion of any kind, but rather an observation that is important to study in the future.

• The methods does a good job of breaking down the timeline a little better and expressing that the questionnaires had a purpose within the longitudinal timeframe, and also states why these different questionnaires were needed (rather than redoing the same questionnaires over and over.

7. PLOS authors have the option to publish the peer review history of their article (what does this mean?). If published, this will include your full peer review and any attached files.

Reviewer #1: No

---

## [Editor Report · Acceptance letter]

12 Feb 2024

PONE-D-23-24545R1 

PLOS ONE

Dear Dr. Drinkwater, 

I'm pleased to inform you that your manuscript has been deemed suitable for publication in PLOS ONE. Congratulations! Your manuscript is now being handed over to our production team.

Kind regards, 

on behalf of

Dr. Giulia Prete 

Academic Editor

PLOS ONE